# Functional Connectivity Biomarker Extraction for Schizophrenia Based on Energy Landscape Machine Learning Techniques

**DOI:** 10.3390/s24237742

**Published:** 2024-12-04

**Authors:** Janerra D. Allen, Sravani Varanasi, Fei Han, L. Elliot Hong, Fow-Sen Choa

**Affiliations:** 1Department of Computer Science and Electrical Engineering, University of Maryland, Baltimore County, Baltimore, MD 21250, USA; sravani1@umbc.edu (S.V.); choa@umbc.edu (F.-S.C.); 2The Hilltop Institute, University of Maryland, Baltimore County, Baltimore, MD 21250, USA; feihan@umbc.edu; 3Department of Psychiatry, University of Texas Health Science Center, Houston, TX 77030, USA; l.elliot.hong@uth.tmc.edu

**Keywords:** energy landscape, fMRI, functional connectivity, biomarker, schizophrenia

## Abstract

Brain connectivity represents the functional organization of the brain, which is an important indicator for evaluating neuropsychiatric disorders and treatment effects. Schizophrenia is associated with impaired functional connectivity but characterizing the complex abnormality patterns has been challenging. In this work, we used resting-state functional magnetic resonance imaging (fMRI) data to measure functional connectivity between 55 schizophrenia patients and 63 healthy controls across 246 regions of interest (ROIs) and extracted the disease-related connectivity patterns using energy landscape (EL) analysis. EL analysis captures the complexity of brain function in schizophrenia by focusing on functional brain state stability and region-specific dynamics. Age, sex, and smoker demographics between patients and controls were not significantly different. However, significant patient and control differences were found for the brief psychiatric rating scale (BPRS), auditory perceptual trait and state (APTS), visual perceptual trait and state (VPTS), working memory score, and processing speed score. We found that the brains of individuals with schizophrenia have abnormal energy landscape patterns between the right and left rostral lingual gyrus, and between the left lateral and orbital area in 12/47 regions. The results demonstrate the potential of the proposed imaging analysis workflow to identify potential connectivity biomarkers by indexing specific clinical features in schizophrenia patients.

## 1. Introduction

Approximately 3 million adults in the United States have schizophrenia, a chronic and severe neurological disorder involving changes in cognition, emotion, and behavior that affect how a person thinks, feels, and behaves [1]. Impaired functional connectivity in schizophrenia underlies neural deficits such as cognitive and psychotic dysfunction. This can be characterized as breakdowns in perceptions, delusions, and organized thoughts, such as the inability to sort and interpret information [2,3]. Scientists have utilized neuroimaging modalities, such as functional magnetic resonance imaging (fMRI), as potential biomarkers to measure spontaneous brain activity and identify functionally involved brain regions. This is limited to static connectivity, as these traditional neuroimaging techniques do not capture the neural dynamic fluctuation(s) exhibited in schizophrenia. While the aim of identifying biomarkers for schizophrenia is to enhance diagnosis, deepen the understanding of its pathophysiology, predict treatment response, and track disease progression, no definitive or reliable methods have yet been universally accepted or integrated into clinical practice. Limitations of neuroimaging biomarkers for schizophrenia include low specificity (for schizophrenia compared to other disorders), methodological variability, and patient heterogeneity [4]. Multimodal approaches that integrate neuroimaging, demographics, and cognitive data help create a more comprehensive and robust biomarker profile.

Characterizing large-scale brain dynamics of resting-state networks can be achieved by analyzing the ROI-to-ROI or seed-based interactions obtained from the resting-state fMRI activity time series. The complex nature of the brain requires a multimodal approach to model brain behavior and function as measuring ROI-to-ROI seed-based connectivity for each individual subject poses a multiple-comparison problem [5]. Therefore, it is important to focus on specific ROIs or clusters sharing similar effects, as defined in the data, while controlling for false positives [5]. Patterns of functional connectivity can be measured using brain atlases made up of *N* ROIs based on their functional and anatomical features. Data-driven hierarchical clustering procedures based on these atlases use ROI-to-ROI anatomical proximity and functional similarity metrics to group related ROIs.

Energy landscape (EL) analysis is an emerging technique that is data-driven and can retrieve brain dynamic information of state transitions [6,7]. Relatively stable and dominant brain activity patterns in high-dimensional neural data, such as fMRI, are observed without any *a priori* behavioral information. EL analysis uses hierarchical clustering to identify the statistical differences between two or more groups and determine the state transitions in a dynamic system. This method is rooted in statistical physics theory, particularly the Ising model, equivalent to the pairwise maximum entropy model (MEM) and the Boltzmann machine [6,7,8]. The EL is achieved by data binarization and calculating the energy and likelihood functions estimated from the Boltzmann distribution.

EL analysis helps determine the brain’s complex activity patterns by examining the stability and transitions between different neural states. Modeling brain activity transitions between attractor or low-energy stable states allows EL analysis to capture temporal dynamics of brain function. Also, EL analysis quantifies the stability of brain states among heterogenous structural and anatomical features. Dynamic patterns of activity can be observed from rs-fMRI data to identify the spatial and temporal patterns that may underlie specific cognitive processes and behaviors [9]. Linking brain activity to behavior dynamics can help discover how dynamic patterns of neural activity produce complex thoughts and behaviors [10,11]. Brain behavior associations are related to functional segregation in coupled clusters, and EL analysis is relevant for identifying the causal relationships of brain function and can contribute to the understanding of cognitive functions and symptoms related to neurological disorders [6,7].

Model- and data-driven approaches could support future treatment development for modifying connectivity. The goal of this study is to identify functional connectivity energy landscape patterns that would differentiate schizophrenic patients and healthy controls and maybe develop into valid biomarkers. This work analyzed resting-state fMRI data for 55 schizophrenia patients and 63 healthy controls using a multimodal imaging analysis workflow and functional resting-state fMRI data. This study applied ROI-to-ROI analysis using the Brainnetome atlas to explore the brain-wide functional connectome differences between both groups. The analysis included finding commonalities and the heterogeneity of functional connectivity patterns across subjects and experimental conditions. To control the family-wise error rate, parametric statistics based on functional network connectivity and nonparametric statistics based on spatial pairwise clustering or permutation/randomization analyses were performed. EL analysis was then used to analyze rs-fMRI data between schizophrenia patients and healthy controls by quantifying the statistical associations of spatially distinct and temporally correlated brain regions with the EL analysis and characterizing large-scale brain dynamics of specific ROIs/clusters.

## 2. Materials and Methods

### 2.1. Data Characteristics

Participants: Fifty-five schizophrenia patients and sixty-three healthy normal controls were included in this study. Using Cochran’s sample size formula, we confirmed that the sample size was sufficient for this study. Patients were recruited from the Maryland Psychiatric Research Center outpatient clinics and several neighboring mental health clinics in the Baltimore area. Controls were recruited from local media advertisements. Diagnoses in patients and the absence of current DSM-IV or V Axis I diagnoses in controls were confirmed by the Structured Clinical Interview for DSM-IV or V. The exclusion criteria were major medical and neurological illnesses, head injury, and substance dependence or substance abuse (except nicotine). All subjects gave their written informed consent approved by the local Institutional Review Board. Overall clinical symptoms were assessed by the 20-item brief psychiatric rating scale (BPRS). The demographic and clinical characteristics for the participants are listed in Table 1.

We used the auditory perceptual trait and state (APTS) and the visual perceptual trait and state (VPTS) scales to measure the corresponding perceptual abnormalities (https://pubmed.ncbi.nlm.nih.gov/29036430/ (accessed on 19 August 2024)). The anomalies are rated for “trait”, defined as longitudinally experienced symptoms over one’s lifetime, and “state”, defined as symptoms recently experienced in the past week. The full scale is available at http://www.mdbrain.org/APTS.pdf (accessed on 19 August 2024). The APTS and VPST scales are self-rated. The brief psychiatric rating scale (BPRS) was used to rate overall symptoms.

### 2.2. Data Acquisition

The resting-state data used for this study were collected by the Maryland Psychiatric Research Center at the University of Maryland School of Medicine. Each participant was instructed to relax and remain motionless with their eyes open. Resting-state fMRI scans were obtained using a 3T Siemens Prisma scanner equipped with a 64-channel head coil. The Human Connectome Project multi-band sequences were used: echo-planar pulse sequence with 2 mm isotropic, TR = 720 ms, MB = 8, AP/PA encoding, and 4 runs × 5.4 min each = 24 min). High-resolution (0.8 × 0.8 × 0.8 mm^3^) T1-weighted MPRAGE images were acquired.

### 2.3. Data Preprocessing and Denoising

The results presented in this manuscript were derived from analyses conducted using CONN release 22.a and SPM release 12.7771 [12,13,14]. Both functional and structural images underwent preprocessing using a flexible pipeline in SPM and were analyzed with CONN, a MATLAB-based software platform for computing, visualizing, and analyzing functional connectivity in fMRI data [12,13,14]. The preprocessing steps included realignment, slice-timing correction, outlier detection, segmentation, MNI space normalization, and smoothing [5]. Functional data were realigned using the SPM12 realign and unwarp procedure [15]. All scans were co-registered and resampled to the reference image (the first scan of the first session) using a least squares approach with a 6-parameter rigid body transformation, followed by b-spline interpolation to correct for motion and magnetic susceptibility effects [15,16]. Temporal misalignment between slices acquired in interleaved Siemens resting-state order was corrected using the SPM12 slice-timing correction procedure, which applied sinc temporal interpolation to resample each slice’s BOLD time-series to a common mid-acquisition time [17,18]. Outlier scans were identified using the Artefact Detection Tools (ARTs) toolbox (https://www.nitrc.org/projects/artifact_detect/ (accessed on 1 July 2024)), flagging scans with global BOLD signal deviations exceeding five standard deviations or with framewise displacement above 0.9. Subsequently, both functional and anatomical data were normalized to MNI space, resampled to 2 mm isotropic voxels, and segmented into gray matter, white matter, and CSF tissue classes using SPM’s unified segmentation and normalization procedures [19,20,21,22]. These procedures separately applied the segmentation and normalization to functional data (using the mean BOLD signal as the reference image) and structural data (using the raw T1-weighted volume as the reference image). Both data types were resampled to a 180 × 216 × 180 mm bounding box with 2 mm isotropic voxels for functional data and 1 mm isotropic voxels for anatomical data, using fourth-order spline interpolation. Finally, functional data were smoothed with an 8 mm full-width half-maximum (FWHM) Gaussian kernel.

For denoising, a standard pipeline was used, which involved regressing out potential confounders, including white matter time series (5 CompCor components), CSF time series (5 CompCor components), motion parameters and their first-order derivatives (12 factors: 6 motion parameters + 6 derivatives), outlier scans (identified as invalid based on a threshold of 98), session effects and their first-order derivatives (2 factors), and linear trends (2 factors) from each functional run [5,23,24]. Afterward, the fMRI BOLD time-series data were bandpass-filtered at 0.01–0.1 Hz [25]. CompCor was used to estimate noise components in white matter and CSF by computing the average BOLD signal and the principal components orthogonal to it, along with motion parameters and outlier scans from each subject’s eroded segmentation masks [26,27]. Based on the number of noise components included in the denoising process, the effective degrees of freedom for the BOLD signal were estimated to range from 246.2 to 301, with an average of 293, across all subjects [19]. As the temporal and spatial domains of connectivity are of primary interest, the time-series data extracted from the sliced fMRI images were utilized for analysis.

### 2.4. First-Level Analysis

A total of 246 ROI seeds, with predefined shapes and locations based on the Brainnetome atlas, a connectivity-based parcellation framework that includes both anatomical and functional connection data, were adjusted for each volume [28]. In the first-level analysis, seed-to-voxel and region-of-interest (ROI-to-ROI) connectivity measures were computed for each subject. ROI-to-ROI connectivity matrices were estimated to characterize the patterns of functional connectivity. The strength of functional connectivity was quantified using Fisher-transformed bivariate correlation coefficients derived from a weighted general linear model (weighted-GLM), which was applied separately to each pair of seed and target regions to model the relationship between their BOLD signal time series [5]. The GLM is a multivariate regression model that evaluates the associations between predictors and outcome variables. Individual scans were weighted using a boxcar signal corresponding to the resting-state condition, which was convolved with an SPM canonical hemodynamic response function and rectified.

### 2.5. Second-Level (Group-Level) Analysis

The second-level analysis assessed cluster-level connectivity between groups using the GLM. Approaches such as functional network connectivity (FNC) and spatial pairwise connectivity (SPC) were employed to define and make inferences about these groups or clusters of connections [5,29]. For FNC, cluster-level inferences in the ROI-to-ROI analyses used the GLM to examine connectivity differences between two groups across all possible pairs of ROIs. For SPC, cluster-level inferences in the ROI-to-ROI analyses employed a randomization/permutation approach to assess connectivity differences between the groups across all ROI pairs. When working with an atlas that includes hundreds of regions and tens of thousands of connections, focusing on groups of neighboring or related connections that share similar effects is a practical approach. Between-subject contrasts (Control = −1, Patient = 1) were applied in the ROI analysis to highlight mean functional connectivity differences between groups. The results from the ROI-to-ROI connectivity measures illustrate the connectivity patterns between all pairs of ROIs within the predefined region set, enabling the simultaneous study of the entire network of connections.

#### 2.5.1. Functional Network Connectivity Analysis

FNC employs multivariate statistics to analyze groups of related ROIs, defined using a data-driven hierarchical clustering method called complete-linkage clustering. This technique groups ROIs based on anatomical proximity and functional similarity, measuring the distance between clusters by the farthest pair of elements (one from each cluster). FNC then examines the entire set of ROI-to-ROI connections, distinguishing within- and between-network connectivity. It calculates an F-statistic for each ROI pair, along with uncorrected and FDR-corrected cluster-level *p*-values. The results demonstrate significant clusters with a familywise corrected *p*-FDR < 0.05 cluster-size threshold [30]. The mean functional connectivity contrasts between groups (Control: −1 vs. Patient: 1) are visualized as a connectome ring, highlighting significant ROI-to-ROI correlations across nodes and networks.

#### 2.5.2. Spatial Pairwise Connectivity Analysis

SPC utilizes optimal leaf ordering to improve hierarchical clustering by minimizing within-cluster variance and arranging leaves to maximize the similarity between adjacent ones. It begins with the full ROI-to-ROI matrix of T- or F-statistics estimated via the GLM, forming a two-dimensional statistical parametric map based on a predefined “height” threshold. Non-overlapping clusters are then identified and characterized by their mass, which is compared to the expected distribution of cluster mass values under the null hypothesis. Iteratively, new parametric maps of T- or F-values are generated using the same threshold, and the properties of the resulting suprathreshold clusters are combined to estimate the probability density under the null hypothesis for the chosen cluster metric. SPC results include uncorrected cluster-level *p*-values, FWE-corrected *p*-values, and FDR-corrected *p*-values, with significant clusters defined by a *p*-uncorrected < 0.05 height threshold and a familywise corrected *p*-FDR < 0.05 cluster-level threshold [30]. Functional connectivity contrasts between groups (Control: −1 vs. Patient: 1) are visualized as a connectome ring, depicting significant ROI-to-ROI correlations across nodes and networks.

### 2.6. Statistical Tests

SPC produced 7/1159 total number of possible clusters. Bonferroni correction, a multiple-comparison correction test, reduces the chances of obtaining false-positive results or type I errors when multiple pairwise tests are performed on a single set of data. The *p*-value for each test must be equal to its alpha, 0.05, divided by the number of tests performed. Bonferroni correction was used to reduce the number of ROIs, using a *p*-value with an alpha of 0.05 and the number of tests corresponding to the cluster-level SPC inference (total number of possible clusters, 1159). After Bonferroni correction, SPC revealed 8 ROIs, derived from 2 clusters, containing significant connectivity between nodes, as shown in Table 2.

### 2.7. REX

The rs-fMRI time-series data were extracted for each subject, spatially aligning the 246-region Brainnetome atlas with the denoised and post-processed rs-fMRI images retrieved from CONN, using REX, a MATLAB-based toolkit used to extract mean image values and time series within the ROIs. REX produced a matrix (888 × 246) of the time-series data and 246-region Brainnetome atlas. In addition to data extraction, REX performs ROI-based analyses of functional data complementing SPM voxel-based analyses. Focusing on the 8 ROIs derived from the post-Bonferroni-corrected SPC cluster-level inference showing the greatest patient and control differences, the original matrix (246 × 888) was transposed to isolate the 8 ROIs to produce a matrix (8 × 888) of the 8 ROIs and the time series.

### 2.8. EL Analysis

The EL analysis toolbox interpreted the matrix (8 × 888) of the 8 ROIs and the time series to measure brain activation of individual connections among the 8 ROIs. Energy landscape analysis is the mapping of possible states in a system. More specifically, energy landscape analysis maps spatial positions of molecules in a system and their corresponding energy levels. It is based on the pairwise MEM, seeking to fit relatively simple second-order models to empirical data. The goal is to identify the trajectory that maps these dynamics due to fluctuations in the brain. Estimating energy landscapes of resting-state activity using fMRI contributes to the understanding of brain dynamics by estimating energy activation. The dynamics from multivariate time-series data can be mapped to the movement of a ‘ball’ constrained on an energy landscape. The pairwise MEM is fitted to analyze probability distributions and energy values of the states generated. This was interpreted based on the properties and functions of the specified (8) ROIs. The energy landscape is achieved by binarizing the data and calculating the energy, likelihood, and maximum likelihood functions. First, the ROIs (*N* = 8) are specified and the fMRI signals are measured at these 8 ROIs. The fMRI signal at each ROI and each time point is binarized into active or inactive. The normalized frequency is computed for each binarized activity pattern. The pairwise maximum entropy model is fitted to the empirical distribution of the 2^N^ activity patterns. For every *N* ROIs, there are 2^N^ possible connectivity states. Energy values and probability distributions are also obtained for each activity pattern. Energy and probability have an inverse relationship showing that small energy values and a larger probability represent states that appear more frequently and are important for representing brain activation.

### 2.9. Paired States

Each subject has 2^N^ states with corresponding energy and probability values. The mean empirical probabilities for each subject were calculated and evaluated at probability rates including 5%, values less than 5%, and values more than 5%. These usage rates represent the states visited. After finding the total number of state pairs that only appear in patients, pairs that only appear in controls, and shared pairs between patients and controls, two-sample *t*-tests of the energies were performed to identify control and patient differences. Complementary paired states were identified based on the empirical probability distribution for each subgroup. The complementary paired states represent connectivity states that are visited more frequently and have lower energy. Bonferroni correction was performed again, separately to reduce the number of states retrieved from energy landscape analysis and identified paired states.

### 2.10. Bivariate Correlation

CONN utilizes the Fisher-transformed correlation coefficients, which are the inverse hyperbolic tangent values of the correlation coefficient. This transformation is expected to improve the normality of the data, therefore making subsequent statistics a little more robust. The Fisher-transformed correlation coefficient values were used for second-level analyses which take individual effects within each subject group and within each of the clusters where there are significant between-group differences. Individual connectivity values were retrieved for each subject, also using Fisher-transformed coefficient values, as second-level covariates. The effects of connectivity strength, paired activation states, and behavioral performance were evaluated to show their linear relationships. Using the R 4.4.2 software, bivariate correlation analysis was conducted, by averaging the complementary states of the paired probabilities, to show subject group contrasts based on connectivity strength, paired activation states, and behavioral performance.

## 3. Results

### 3.1. Neuropsychological Assessment

Two-sample *t*-tests were conducted, assuming equal variances for the age, memory, and processing speed tests. Two-sample *t*-tests were conducted, assuming unequal variances for the brief psychiatric rating scale (BPRS), auditory perceptual trait and state (APTS), visual perceptual trait and state (VPTS), working memory score, and processing speed score. Chi-square tests were conducted on the sex and smoker/non-smoker variables. The chi-square calculated value is significant when equal to or more than the chi-square critical value. The null hypothesis (H_0_) is rejected if the chi-square calculated value is greater than the chi-square critical value. The *p*-value can also determine whether the null hypothesis must be accepted or rejected, such that if *p*-value ≤ α, then the null hypothesis is rejected, or if *p*-value > α, the null hypothesis is accepted. For the age variable, the chi-square calculated value is less than the chi-square critical value or 0.079 < 3.84. Also, the *p*-value is greater than alpha or 0.78 > 0.05, so the alternative hypothesis is rejected and the null hypothesis is accepted in both instances. For the sex variable, the chi-square calculated value is less than the chi-square critical value or 0.077 < 3.84. Also, the *p*-value is greater than alpha or 0.78 > 0.05, so similarly to the age variable, the results are not significant.

### 3.2. Spatial Pairwise Connectivity Maps

The analysis conducted on ROI-to-ROI connectivity during rest found no significant results between smoker and non-smoker subjects and male and female subjects. Figure 1a shows the significant average connectivity differences between control and patient subjects. Significant connections between the nodes are represented as a connectome ring. Clusters represent a high concentration of positively connected or correlated nodes [31].

### 3.3. Reduced ROIs

Figure 1b illustrates the reduced ROIs identified using the SPC method, following post-Bonferroni correction. These ROIs include the left and right ventromedial parieto-occipital sulcus (vmPOS) and the left and right rostral lingual gyri (rLinG), which are part of the medioventral occipital cortex within the occipital lobe. These regions are part of the visual network. The left area 41/42 (A41/42) belonged to the superior temporal gyrus of the temporal lobe and was part of the auditory and attention networks. The left lateral and orbital areas 12/47 (A1247l |A12/47o) belonged to the orbital gyrus of the frontal lobe and are not defined by any brain network. Lastly, the left area 1/2/3 of the tongue and larynx belonged to the precentral gyrus of the frontal lobe and was part of the sensorimotor network. The AAL atlas was used to subdivide the ROIs into the corresponding networks.

### 3.4. Activation Maps of Paired States Obtained from EL Analysis

Individual ROI Analysis: There were 2^N^ or 256 possible connectivity states across the ROIs. After Bonferroni correction, four significant paired activation states were found. In Figure 2, the activation maps of the paired states indicate the ROIs that were active or inactive together, and thus correlated or anti-correlated. The boxplots of the four significant paired activation states are shown in Figure 3, showing that the HC group has slightly higher frequencies than the SSD group, suggesting more stable connections.

### 3.5. Physiological Meaning of Paired States

Paired states 23 and 234 have different activation activity in the left ventromedial parieto-occipital sulcus, right rostral lingual gyrus, left area 41/42, and left lateral area 12/47 regions, to paired states 123 and 134. The left ventromedial parieto-occipital sulcus and right rostral lingual gyrus both belong to the occipital lobe of the medioventral occipital cortex, suggesting that the orbitofrontal visual cortex is involved in visual and auditory hallucinations. This effect is on the first pair of states, but not on the second pair. Visual hallucinations are a misinterpretation of internally generated imageries as external, which normally could be regulated or ‘suppressed’ by higher executive areas such as the prefrontal cortex. A connectivity problem that controls this process may lead to the misperception of one’s own thoughts, which could explain the hallucination generations.

## 4. Discussion

In this work, EL analysis was used to compare the dynamics between healthy normal populations and patients with schizophrenia spectrum disorders. The ROIs were calculated along the rs-fMRI time series, and the statistical differences of the energy values were analyzed. The interactions of eight ROIs were analyzed and EL analysis was used to extract connectivity signature states from the resting-state fMRI data of the schizophrenia spectrum disorder and healthy control groups. Relying on single modalities, such as structural or functional MRI, has been limiting and does not capture the full complexity of brain function. EL analysis can integrate information from structural and functional MRI to help develop more comprehensive and robust multimodal biomarkers for schizophrenia.

The observed differences in the eight identified brain regions such as the ventromedial parieto-occipital sulcus, lingual gyri, Brodmann areas 41/42, lateral/orbital areas 12/47, and the primary somatosensory areas for the tongue and larynx in schizophrenia point to structural and functional abnormalities in these regions. These changes may result from disrupted neurodevelopmental processes, leading to impaired connectivity and altered cortical morphology. Additionally, these differences are influenced by a combination of genetic predispositions and environmental factors, which collectively shape brain structure and connectivity in schizophrenia. This complex interplay underscores the multifactorial nature of the disorder, requiring an integrative approach to better understand its pathology. Schizophrenia is associated with reductions in gray matter volume (GMV) and functional connectivity density in posterior cortical regions, including the parieto-occipital sulcus and lingual gyri. These regions are linked to processing vision and encoding visual memories. These structural and functional abnormalities may impair visual–spatial processing and sensory integration, disrupting the brain’s ability to maintain sensory coherence—a cognitive function frequently compromised in individuals with schizophrenia [32,33]. The primary auditory cortex, encompassing Brodmann areas 41/42, exhibits both structural reductions and functional disruptions in individuals with schizophrenia. These abnormalities are strongly associated with auditory hallucinations and deficits in auditory processing—key features of the disorder. Impaired cortical thickness and dysfunctional connectivity within this region contribute to difficulties in processing and orienting auditory stimuli, highlighting its role in the sensory and cognitive symptoms observed in schizophrenia [33]. The lateral and orbital regions of the prefrontal cortex in schizophrenia often show reduced GMV and abnormal activity patterns. These regions are linked to language processing and comprehension, but are also critical for higher-order cognitive functions such as decision-making, emotional regulation, and social cognition. Dysfunction in these areas contributes to the cognitive impairments and emotional dysregulation commonly observed in schizophrenia, highlighting their role in the disorder’s complex symptomatology [32].

The eight identified ROIs showed significant SSD and HC differences. The abnormal energy landscape findings were significantly associated with the severity of the auditory and visual perceptual disturbances in the SSD group. The two-sample *t*-tests measured the significance of the observations and found complementary connectivity state pairs, which indicated the frequency of the brain visiting these states. These paired states form deeper basins in the energy landscape, resulting in higher stability. This shows that neuron firing frequency is related to activation energy. Alterations in the primary somatosensory cortex may indicate disruptions in sensory integration, potentially contributing to deficits in bodily self-awareness or the occurrence of tactile hallucinations. However, this connection remains underexplored compared to other brain regions affected in schizophrenia, necessitating further research to clarify its role in the disorder’s sensory and perceptual abnormalities [32,33].

This work shows that the schizophrenia and healthy control groups have significant differences in these paired states along these ROIs. The activation among the ROIs is weaker for schizophrenia, as suggested by the findings that the probability of being at particular states is lower compared to that of healthy controls in these ROIs. Paired states were identified as potential biomarkers between patients and controls. Measuring the probability distributions of these signature states helps to better understand brain region structures and functions. Quantitative measurement of brain connectivity and signature state extraction are important for identifying patients with neuropsychiatric disorders. EL analysis can identify network- or region-specific energy patterns relative to schizophrenia. Identifying the stability or instability of attractor states could help clinicians develop interventional strategies to reshape the patient’s brain dynamics. If further validated, this quantitative method could be developed into a biomarker(s) to guide future treatment development for modifying connectivity and improving brain function.

### 4.1. Advantages

EL analysis can help identify which states are either robust (low-energy) or transient (high-energy). Transient states could be linked to cognitive processes and symptoms contributing to brain function. EL analysis offers insights into the underlying mechanisms of schizophrenia, helping us to understand how brain structure affects functional dynamics.

### 4.2. Limitations

In this work, the number of regions was reduced to account for the computational cost of the EL analysis. This possibly yielded oversimplified or missed brain dynamic information. The temporal resolution of fMRI is lower compared to many neural processes; therefore, capturing the rapid transitions between brain states has been difficult. Variability in preprocessing steps and assumptions about the system can influence the results and affect reproducibility. It is still challenging to interpret the attractor stability or transition probabilities based on neuropsychological or cognitive mechanisms; therefore, their clinical relevance is still in research phase. Brain dynamics varies across individuals, so applying group-level analyses to individual subject analyses may influence biomarker development.

To enhance the reliability, sensitivity, and interpretability of fMRI studies, future research must focus on refining study design parameters. Group stratification based on factors such as symptom severity can help reduce variability and uncover group-specific effects. Employing higher spatial and temporal resolution in imaging protocols will allow for the detection of finer structural and functional details of brain activity. Integrating task-based and resting-state fMRI can offer complementary perspectives—task-based fMRI is well suited for examining specific cognitive processes, while resting-state fMRI reveals intrinsic connectivity patterns. Analyzing dynamic, time-varying connectivity rather than static measures can provide deeper insights into the fluctuating dynamics of brain networks. Furthermore, combining fMRI with other complementary techniques, such as diffusion tensor imaging (DTI), electroencephalography (EEG), or magnetoencephalography (MEG), can yield a more comprehensive understanding of the relationships between brain structure and function.

## Figures and Tables

**Figure 1 sensors-24-07742-f001:**
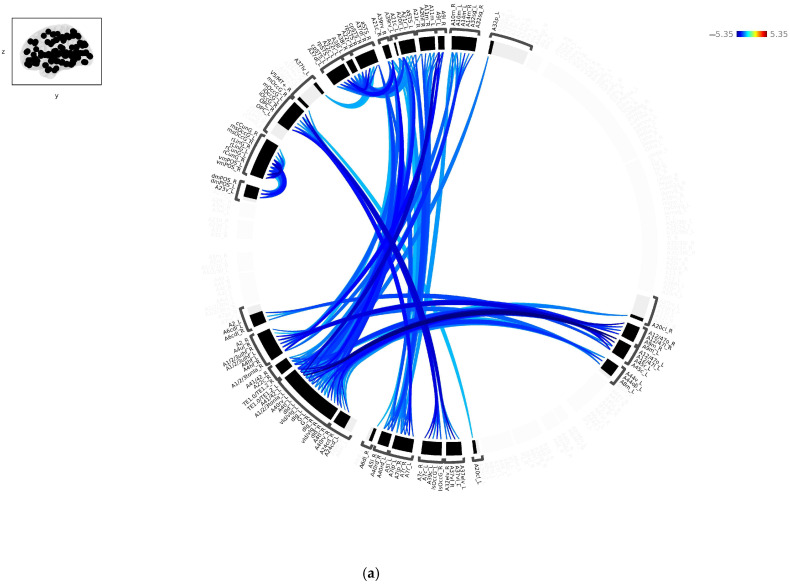
The mean FC for the between-subject [Control (−1) vs. Patient (1)] contrast presented as a connectome ring showing nodes with a decreased FC for patients. The connectome ring network of correlations, (**a**) before and (**b**) after Bonferroni correction, showing the reduced number of ROIs in a single brain display computed using spatial pairwise connectivity (SPC).

**Figure 2 sensors-24-07742-f002:**
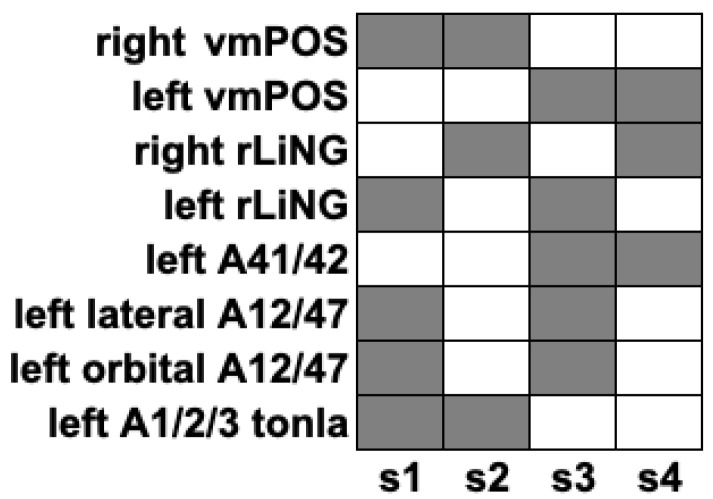
The four activation states of the HC and SSD groups.

**Figure 3 sensors-24-07742-f003:**
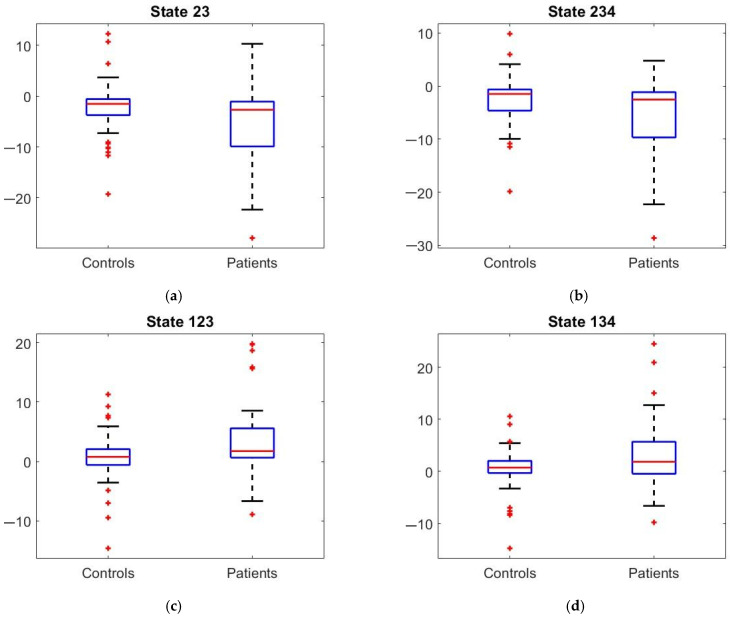
The dynamic properties of the four significant paired activation states. The mean frequency of the brain states for the HC and SSD groups. Complementary energy states 23 and 234 and 123 and 134 were identified as potential connectivity biomarkers.

**Table 1 sensors-24-07742-t001:** Demographics of HN controls and SSD patients.

	HN Controls *N* = 63	SSD Patients *N* = 55	Differences
Age	36.75 ± 13.62	34.53 ± 13.13	HN vs. SSD: t = 0.9, *p* = 0.37
Sex	42 (Male) 21 (Female)	38 (Male) 17 (Female)	HN vs. SSD: *χ*2 = 0.079, *p* = 0.78
Smoker	18 (Smoker) 42 (Non-Smoker)	17 (Smoker) 38 (Non-Smoker)	HN vs. SSD: *χ*2 = 0.077, *p* = 0.78
BPRS Total	24.1 ± 4.91	37.09 ± 12.39	HN vs. SSD: t = −7.24, *p* < 0.001
APTS—State	0.08 ± 0.22	0.72 ± 0.94	HN vs. SSD: t = −4.94, *p* < 0.001
APTS—Trait	0.42 ± 0.48	1.44 ± 1.23	HN vs. SSD: t = −5.78, *p* < 0.001
VPTS—State	1.28 ± 3.19	14.93 ± 20.89	HN vs. SSD: t = −4.04, *p* < 0.001
VPTS—Trait	0.19 ± 0.24	1 ± 1.09	HN vs. SSD: t = −5.61, *p* < 0.001
Working Memory Score	21.43 ± 4.06	18.55 ± 5.21	HN vs. SSD: t = 3.28, *p* < 0.01
Processing Speed Score	76.54 ± 14.07	62.37 ± 20.51	HN vs. SSD: t = 4.32, *p* < 0.001

HN: healthy normal; SSD: schizophrenia spectrum disorder; BPRS: brief psychiatric rating scale; APTS: auditory perceptual trait and state; VPTS: visual perceptual trait and state. Values are means or frequencies (for sex and smoker) with SDs.

**Table 2 sensors-24-07742-t002:** Statistical cluster analysis of clusters using spatial pairwise connectivity (SPC), presenting significant connectivity between nodes (post-Bonferroni correction).

Analysis		Statistic	*p*-Unc	*p*-FDR
Cluster 1		Score = 97.42	<0.001	0.013
		Mass = 145.42	<0.001	0.012
		Size = 6	<0.001	0.020
Left A1/2/3tonla	Left A12/47o	T(116) = −5.35	<0.001	0.031
Left A1/2/3tonla	Left A12/47l	T(116) = −5.00	<0.001	0.028
Left A41/42	Left A12/47o	T(116) = −4.37	<0.001	0.051
Cluster 2		Score = 77.30	0.006	0.044
		Mass = 124.63	0.002	0.014
		Size = 6	0.003	0.020
Left rLinG	Left vmPOS	T(116) = −4.79	<0.001	0.038
Right rLinG	Left vmPOS	T(116) = −4.48	<0.001	0.051
Left rLinG	Right vmPOS	T(116) = −4.39	<0.001	0.051

*p*-unc: uncorrected *p*-value; *p*-FDR: false discovery rate-adjusted *p*-value.

## Data Availability

The original contributions presented in this study are included in the article. Further inquiries can be directed to the corresponding author.

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
