# Peer review of "Functional Connectivity Biomarker Extraction for Schizophrenia Based on Energy Landscape Machine Learning Techniques"

_sensors, 2024, doi:10.3390/s24237742_

Round 1

Reviewer 1 Report

Comments and Suggestions for Authors

In this work, the authors used energy landscape analysis to compare the dynamics between healthy controls and patients with schizophrenia and found that the brains of individuals with schizophrenia have abnormal energy landscape patterns.

1. When multiple pairwise tests are performed, the authors correctly used multiple comparison correction tests to reduce the chance of obtaining false-positive results.

2. The study included fifty-five schizophrenia patients and sixty-three healthy normal controls. Is the sample size sufficient? Is there sufficient statistical power?

3. Limitations and advantages should be discussed in the article. 

4. Please discuss the possible reasons for the differences found in these eight ROIs between patients and healthy controls, and the mechanisms behind them. Have these differences been identified or discussed in previous studies?

5. Please describe the similarities and differences between this article and the 30th reference -- J. D. Allen, S. Varanasi, E. Hong, and F.-S. Choa, “Energy landscape analysis of fMRI data from Schizophrenic and 585 healthy subjects,” in Signal Processing, Sensor/Information Fusion, and Target Recognition XXX, SPIE, Apr. 2021, pp. 586 258–266. doi: 10.1117/12.2588046.

Author Response

Comment 2: The study included fifty-five schizophrenia patients and sixty-three healthy normal controls. Is the sample size sufficient? Is there sufficient statistical power?

Response: Thank you for pointing this out. Yes, this study sample size is sufficient. Using Cochran’s sample size formula, we confirmed that the sample size is sufficient for this study. We assumed an estimated population proportion of 1.1% of patients with schizophrenia, since that is the population of patients with schizophrenia in the United States. This can be found in the revised manuscript on line 101-102. 

Comment 3: Limitations and advantages should be discussed in the article.

Response: Agree. I have, accordingly, added and emphasized the limitations and advantages of the study.  This can be found in the Discussion section of the revised manuscript on lines 439-468. 

Here is the direct wording from lines 439-443:

Advantages: EL analysis can help identify which states are either robust (low-energy) or transient (high-energy). Transient states could be linked to cognitive processes and symptoms contributing to brain function. EL analysis offers insights into the underlying mechanisms of schizophrenia, helping to understand how brain structure affects functional dynamics.

Here is a summary of lines 444-468:

Limitations: This study reduced the number of brain regions analyzed to manage computational costs, potentially oversimplifying or missing key dynamic information. Limitations such as the low temporal resolution of fMRI, variability in preprocessing, and challenges in linking attractor stability to cognitive mechanisms hinder reproducibility and clinical relevance. Brain dynamics also vary across individuals, complicating biomarker development. To improve reliability and sensitivity, future research should refine study designs, use higher-resolution imaging protocols, and integrate task-based and resting-state fMRI. Exploring dynamic connectivity and combining fMRI with complementary techniques like DTI, EEG, or MEG can provide a more comprehensive understanding of brain structure and function.

Comment 4: Please discuss the possible reasons for the differences found in these eight ROIs between patients and healthy controls, and the mechanisms behind them. Have these differences been identified or discussed in previous studies?

Response: I agree with this and have incorporated your suggestion in the Discussion section on lines 384-411. Please find the summary of the differences in these eight ROIs below:

Here is a summary of lines 444-468:

Schizophrenia is associated with structural and functional abnormalities in eight brain regions, including the parieto-occipital sulcus, lingual gyri, Brodmann areas 41/42, and prefrontal areas 12/47. These changes, influenced by disrupted neurodevelopment, genetic predisposition, and environmental factors, highlight the disorder's multifactorial nature. Reductions in gray matter volume (GMV) and functional connectivity in posterior cortical regions, like the parieto-occipital sulcus and lingual gyri, impair visual-spatial processing and sensory integration, affecting sensory coherence. The primary auditory cortex shows structural and functional disruptions linked to auditory hallucinations and deficits in auditory processing. Additionally, abnormalities in prefrontal regions contribute to language impairments, cognitive deficits, emotional dysregulation, and social dysfunction, underscoring the complexity of schizophrenia's pathology.

Comment 5: Please describe the similarities and differences between this article and the 30th reference -- J. D. Allen, S. Varanasi, E. Hong, and F.-S. Choa, “Energy landscape analysis of fMRI data from Schizophrenic and healthy subjects,” in Signal Processing, Sensor/Information Fusion, and Target Recognition XXX, SPIE, Apr. 2021, pp. 586 258–266. doi: 10.1117/12.2588046.

Response: Thank you for pointing this out. The reference above is a conference paper that contributes to this full paper. I have also revised the introduction to include less of the literature review describing the brain networks responsible for schizophrenia. Also, the methods of the two are different, as in this paper we incorporated the CONN analysis workflow to help us derive the regions of interest that were most significant, rather than focusing primarily on the brain networks. 

Reviewer 2 Report

Comments and Suggestions for Authors

I've attached my comments for the paper as a word-file below.

Comments on the Quality of English Language

The language can be improved by simplifying sentences and removing the un-relevant information. 

Author Response

Comment 1: Main Concerns: 1. In general, the language of the paper can be improved. There’s some unrelated/unimportant information provided now and then in the manuscript. Such as in the abstract you mentioned what kind of multiple comparison corrections are used which I think is not necessary to include in the abstract. And in the first session of results, line 380 you said the p-value if greater than alpha…, which I think it’s enough to say it’s not significant while providing the p-values.

Response: Thank you for this suggestion. The paper was updated to reflect the removal of the unrelated/unimportant information provided, such as the line in the abstract mentioning the multiple corrections used. Also, as noted below in Comment 3, parts of the introduction were removed to focus on the biomarkers.  The first section of the results mentioning the p-value if greater than alpha was updated to simplify the statement (line 319). 

Comment 2: In Abstract 2. The main contribution of this paper is the EL method. But you didn’t touch on any potential advantages of this method compared to the existing method in the abstract. I think it’s better to summarize in one or two sentences of the reason why you applied this method over others.

Response: Thank you for pointing this out. This has been included and can be found in the revised manuscript on line 18-19. Below is the direct wording that was added to the abstract. 

EL analysis captures the complexity of brain function in schizophrenia, by focusing on functional brain state stability and region-specific dynamics.

Comment 3: In Introduction 3. The introduction is too heavy for me, providing a lot of information. I suggest re-structure the introduction as follows: a). Schizophrenia is a common disease all over the world, but still no reliable biomarkers have been developed. b). What biomarkers have been achieved by existing methods and in what sense are these biomarkers not satisfying? c). What additional information can be provided by the EL methods to overcome these difficulties?

Response: I agree with this comment. Therefore I edited the introduction to focus on the points mentioned (lines 36-48 and lines 70-74). 

Here is a summary of lines 36-38:

Scientists use neuroimaging, like fMRI, to study brain activity and identify regions linked to schizophrenia, but traditional techniques are limited by their inability to capture neural dynamics. While biomarkers aim to improve diagnosis, treatment prediction, and understanding of the disease, they remain unreliable due to low specificity, methodological inconsistencies, and patient variability. Multimodal approaches that combine neuroimaging with demographic and cognitive data show promise for developing more robust biomarker profiles.

Here is a summary of lines 70-74:

EL analysis examines the brain's complex activity patterns by modeling transitions between stable neural states (attractors) and capturing temporal dynamics. It also quantifies the stability of brain states in relation to diverse structural and anatomical features.

Comment 4: In Methods 4. The resting-state connectivity has been reported to be influenced by the global signal. If I understand correctly, the global signal was not removed during the preprocessing. What’s the results would be like if it’s removed?

Response: Thank you for this suggestion. It would have been interesting to explore this aspect. However, in the case of our study, it seems slightly out of scope because the Global Signal Regression (GSR) is an alternative approach to the default denoising pipeline that CONN uses. Generally, the default denoising pipeline involves linear regression of potential confounding effects in the BOLD signal and temporal band-pass filtering, whereas GSR uses the average BOLD signal as a potential confounding effect. “GSR is generally not recommended, since it can introduce artifactual biases, remove potentially neural components, and introduce confounding effects across populations”, as noted here: https://web.conn-toolbox.org/fmri-methods/denoising-pipeline

Comment 5: In Results 5. Line 426-427, You reported the p values for the correlation analyses, they both are higher than 0.05. Then why did you still say it’s significant? Am I missing something? Furthermore, the correlations in the Schizophrenia patients in Figure 4 are driven largely by the outlier at the topright corner. What is the result like if you remove this patient?

Response: Thank you for this suggestion. As it turns out, removing the outlier at the top right corner changed the results, therefore this association plot was not strong enough to suggest the relationship between connectivity and perceptual scores. Section 3.5 was updated and section 3.6 was removed to reflect the changes in the article.

Comment 6: Minor Concerns: In methods 1. Line 184, please provide all scanning parameters both for functional and anatomical runs.

Response: Thank you for this suggestion. To my knowledge, the scanning parameters for both functional and anatomical runs are provided. Can you please clarify what additional parameters are needed? 

In addition to the above comments, sections 2.1-2.5 pointed out by the reviewers have been updated and rephrased as there was overlap with other published articles. We look forward to hearing from you in due time regarding our submission and to respond to any further questions and comments you may have. Other sections of the manuscript were updated to explain the relevance of the EL analysis method. 

Here is the direct wording from lines 379-383:

Relying on single modalities, such as structural or functional MRI have been limiting and does not capture the full complexity of brain function. EL analysis can integrate information from structural and functional MRI to help develop more comprehensive and robust multimodal biomarkers for schizophrenia.

Here is the direct wording from lines 418-423:

Alterations in the primary somatosensory cortex may indicate disruptions in sensory integration, potentially contributing to deficits in bodily self-awareness or the occurrence of tactile hallucinations. However, this connection remains underexplored compared to other brain regions affected in schizophrenia, necessitating further research to clarify its role in the disorder's sensory and perceptual abnormalities [32], [33].

Here is the direct wording from lines 432-435:

EL analysis can identify network- or region- specific energy patterns relative to schizophrenia. Identifying the stability or instability of attractor states could help clinicians develop interventional strategies to reshape the patient’s brain dynamics.

Round 2

Reviewer 2 Report

Comments and Suggestions for Authors

I'm OK with the current version.